# Fecal DNA Virome Is Associated with the Development of Colorectal Neoplasia in a Murine Model of Colorectal Cancer

**DOI:** 10.3390/pathogens11040457

**Published:** 2022-04-11

**Authors:** Yingshi Li, Fan Zhang, Huimin Zheng, Sanjna Kalasabail, Chloe Hicks, Ka Yee Fung, Adele Preaudet, Tracy Putoczki, Julia Beretov, Ewan K. A. Millar, Emad El-Omar, Xiao-Tao Jiang, Howard Chi Ho Yim

**Affiliations:** 1UNSW Microbiome Research Centre, St George and Sutherland Clinical Campuses, School of Clinical Medicine, UNSW Medicine and Health, The University of New South Wales, Sydney, NSW 2052, Australia; kiki.2.lee@hotmail.com (Y.L.); fan.zhang7@unsw.edu.au (F.Z.); zhenghuimin91@126.com (H.Z.); s.kalasabail@student.unsw.edu.au (S.K.); chloe.hicks@unsw.edu.au (C.H.); e.el-omar@unsw.edu.au (E.E.-O.); 2Personalised Oncology Division, The Walter and Eliza Hall Institute of Medical Research, Victoria, VIC 3052, Australia; fung.k@wehi.edu.au (K.Y.F.); preaudet.a@wehi.edu.au (A.P.); putoczki.t@wehi.edu.au (T.P.); 3Department of Medical Biology, University of Melbourne, Victoria, VIC 3053, Australia; 4St George and Sutherland Clinical Campuses, School of Clinical Medicine, UNSW Medicine and Health, The University of New South Wales, Sydney, NSW 2052, Australia; julia.beretov@health.nsw.gov.au (J.B.); ewan.millar@health.nsw.gov.au (E.K.A.M.); 5Department of Anatomical Pathology, NSW Health Pathology, St George Hospital, Kogarah, NSW 2217, Australia

**Keywords:** bacteriophage, colorectal neoplasia, virome

## Abstract

Alteration of the gut virome has been associated with colorectal cancer (CRC); however, when and how the alteration takes place has not been studied. Here, we employ a longitudinal study in mice to characterize the gut virome alteration in azoxymethane (AOM)-induced colorectal neoplasia and identify important viruses associated with tumor growth. The number and size of the tumors increased as the mice aged in the AOM treated group, as compared to the control group. Tumors were first observed in the AOM group at week 12. We observed a significantly lower alpha diversity and shift in viral profile when tumors first appeared. In addition, we identified novel viruses from the genera *Brunovirus*, *Hpunavirus* that are positively associated with tumor growth and enriched at a late time point in AOM group, whereas members from *Lubbockvirus* show a negative correlation with tumor growth. Moreover, network analysis revealed two clusters of viruses in the AOM virome, a group that is positively correlated with tumor growth and another that is negatively correlated with tumor growth, all of which are bacteriophages. Our findings suggest that the gut virome changes along with tumor formation and provides strong evidence of a potential role for bacteriophage in the development of colorectal neoplasia.

## 1. Introduction

Colorectal cancer (CRC) is the third most commonly diagnosed cancer and the second most common cause of cancer death [1]. An estimated 1.14 million new CRC cases and 576,858 CRC deaths occurred worldwide in 2020 [1]. Many risk factors have been associated with an increased risk of CRC, including genetics, age, a history of inflammatory bowel disease (IBD), diet, and more recently, the gut microbiome [2,3,4,5,6]. The gut microbiome is constituted of over 100 trillion microorganisms (including bacteria, viruses, archaea, and fungi) and their collective genomes [7]. Patients with CRC harbor a dysbiotic gut microbiota. This includes a higher relative abundance of bacteria suspected to play a role in colorectal carcinogenesis, such as *Fusobacterium* spp., *Bacteroides fragilis*, *Escherichia coli*, *Streptococcus bovis*, and *Enterococcus faecalis* [7]. In particular, enrichment of *Fusobacterium nucleatum* (Fn) was found in colorectal carcinomas using whole-genome sequencing and confirmed by a large-scale study of colorectal cancer tissue samples [8,9]. A reduction in bacterial genera such as *Clostridium* and *Faecalibacterium* that play a protective role was also observed in CRC patients [10]. 

While the association of CRC and bacteriome have been extensively studied, little is known about the role of the gut virome. The gut virome comprises endogenous retroviruses, eukaryotic viruses, and bacteriophages [11]. Alteration of the human gut virome has been associated with diseases such as Type-1 Diabetes, Type-2 Diabetes, IBD, HIV infection, and cancer [12,13,14,15,16,17,18]. Recent studies showed controversial findings about the gut virome in stool samples of CRC patients. A study by Nakatsu et al. showed a relative increase in bacteriophage alpha diversity in CRC patients compared to healthy controls [17], whereas Hannigan et al. did not find any significant differences in either Shannon diversity or viral richness [18]. According to the operational viral units (OVUs) based on relative abundance profiles, Hannigan et al. suggested that the cancer-associated virome mainly consisted of temperate bacteriophages, and the virome signatures were from OVUs belonging to *Siphoviridae* and *Myoviridae* [18]. *Orthobunyavirus* was identified as the most important viral genus that differentiated CRC patients from controls, although its potential role in enteric diseases is yet to be explored [17]. Evidence also suggested a potential disease-stage specific alteration in the enteric virome [17]. However, further studies are needed to validate this finding.

The studies above provided a snapshot of the differences in virome in patients with colorectal neoplasia compared to healthy controls, however, it is unclear whether the virome alteration is a cause or consequence of tumor growth and how the virome changes over time. Thus, we hypothesize that the gut virome changes along with the development of colorectal neoplasia. Here, we tested this hypothesis by setting up a longitudinal study in mice. Stool samples were collected biweekly. We compared the stool virome composition and abundance between mice that were treated with azoxymethane (AOM), a carcinogen commonly used for inducing CRC, and phosphate-buffered saline (PBS) at six time points. We then established correlations between the gut virome and tumor growth. 

## 2. Results

### 2.1. The Number and Size of Tumors Increased as the Mice Aged after Treatments

To investigate changes in the gut virome during the colorectal neoplasm development, a total of 96 samples were collected from eight AOM- and eight PBS-treated mice across six timepoints. The weight of the mice increased as they aged, but no significant difference was found between AOM- and PBS-treated mice (Figure 1a). By using colonoscopy, colorectal tumors were first observed in mice treated with PBS and AOM 12 weeks after injection (Figure 1c,d). One tumor per mouse was found in two out of eight PBS-treated mice at week 12 after injection (Figure 1c,d and Appendix A). One of these tumors disappeared after week 12 of injection while the other tumor did not increase its size throughout the rest of the experiment (Figure 1c and Appendix A). In contrast, the number and size of these tumors AOM-treated mice increased as the mice aged, with a maximum of 17 tumors per mouse observed in AOM-treated mice at week 24 (Appendix A). The normal and tumor tissues identified via colonoscopy were verified by histopathological examination on the biopsies taken at different time points (Figure 1e and Appendix A). Two-way ANOVA test showed a significant difference in tumor size between AOM- and PBS-treated mice from week 12 (*p* < 0.05) (Figure 1b) and in the number of tumors from week 14 (*p* < 0.05) (Figure 1c). These data suggest that AOM induces colorectal neoplasia starting at week 12 after injection and the neoplasia grows when the mice get aged. 

### 2.2. Altered Gut Virome Diversity in Colorectal Neoplasia When Tumors First Appeared

We first explored the influence of colorectal neoplasia on gut virome diversity by evaluating the differences in virome between treatment and time, using the observed number of OVUs, Shannon diversity, and Bray–Curtis metrics. No significant decrease in the observed number of OVUs was observed in AOM treated mice among different time points (Figure 2a). However, a significant decrease in the observed number of OVUs was observed in PBS treated mice at week 20 compared to weeks 0 (*p*-value 0.001, *Q* value 0.002) and 12 (*p*-value 0.007, *Q* value 0.009). These patterns were not observed in AOM treated mice (Appendix A). A lower Shannon diversity was observed in AOM models at week 12 compared to weeks 10 (*p*-value 0.021, *Q* value 0.083) and 0 (*p*-value 0.028, *Q* value 0.225) (Figure 2b). In PBS treated mice, a significant lower Shannon diversity was observed at week 20 compared to weeks 0 (*p*-value 0.0002, *Q* value 0.001) and 12 (*p*-value 0.015, *Q* value 0.017), and at week 14 compared to weeks 0 (*p*-value 0.003, *Q* value 0.019) and 12 (*p*-value 0.05, *Q* value 0.133) (Appendix A). These data suggest that aging may contribute to the decreased Shannon diversity, while the occurrence of the first tumor is linked to the early onset of this decreased Shannon diversity. 

A significant shift in the viral profile was also observed at weeks 12, 14, and 24 compared to week 0 in AOM models (PERMANOVA, *p*-value 0.001, *Q* value 0.066) (Figure 3). AOM virome at week 0 shifted away from that at week 12 but returned to where it started at weeks 20 and 24, although the change in beta diversity was not significant (Appendix A). No significant shift in viral profile was observed in PBS treated mice (*Q* value cut off 0.1). These data suggest that gut virome diversity has been altered since the first tumor appeared. 

### 2.3. Gut Virome Composition and Viruses Enriched in Colorectal Neoplasia

Because over 55% of the OVUs are unclassified, virome composition analysis was done on classified OVUs only. At the family level, 76.51 ± 4.21% of the OVUs were unclassified; the OVUs classified at the family level were dominated by bacteriophages from *Siphoviridae* (13.56 ± 3.42%) and *Myoviridae* (9.59 ± 1.24%) (Figure 4a). At the genus level, 96.4 ± 0.8% of the OVUs were unclassified (Figure 4b). Omitting the OVUs that were unclassified at the genus level, members of *Lagaffevirus* were the most abundant (1.57 ± 0.41%), followed by *Brunovirus* (0.94 ± 1.13%) and *Toutatisvirus* (0.47 ± 0.23%) (Appendix A). *Brunovirus* had a surge in abundance in week 12 in both AOM and PBS treated mice, however, this change in abundance was not significant (Dunn’s test, *p* > 0.05) (Appendix A). No OVUs were annotated at the species level. Although this virome composition came from classified OVUs, there was still a significant proportion of unclassified viruses at the family and genus levels due to many viruses being poorly characterized taxonomically.

To determine whether any viruses at the OVU level and genus level were enriched or depleted in the AOM model and PBS model, we used the LEfSe algorithm to compare the gut virome composition of the AOM treated mice to that of the PBS treated mice at each timepoint. At the genus level, *Toutatisvirus* was enriched at week 10 in PBS treated mice (Figure 4c). *Brunovirus* and *Hpunavirus* were enriched in AOM models at week 24, while *Fromanvirus* was enriched in PBS models at week 24 (Figure 4d). The OVUs and viral genera that were differentially abundant in AOM models may be associated with the tumor virome. No viral genera were consistently enriched in more than 1 timepoint. No differentially abundant genera were identified in weeks 0, 12, 14, and 20.

At the OVU level, two OVUs were enriched in three or more time points in the AOM group. OVU#200524 was enriched in weeks 10, 12, 14, and 24, while OVU#208593 was enriched in weeks 0, 10, and 20 (Table 1). Conversely, four OVUs were consistently enriched in the control group, including OVU#272093, OVU#79336, OVU#175582, and OVU#99458 (Table 1). Unlike most of the OVUs that were enriched in early time points (weeks 0, 10, and 12), OVU#99458 was enriched at the later time points (weeks 14, 20, and 24). The full list of OVUs that were differentially abundant in either of the groups is listed in Appendix A.

### 2.4. Association of the Gut Virome with Tumor Growth

To investigate the correlation between viruses and tumor characteristics, correlation analysis was performed with R package MaAsLin2. MaAsLin2 identified 8 highly abundant OVUs that had a significant positive correlation with weeks after injection in AOM treated mice (*Q* ≤ 0.05) (Figure 5a). Out of the 8 OVUs, 7 were unclassified and 1 belonged to the genus *Brunovirus* (OVU#24213). None of these OVUs were significantly correlated with time in PBS treated mice (Appendix A). Only 1 OVU showed a significant correlation with the number of tumors—this OVU did not show a significant correlation with time and is unclassified (Figure 5b). Three of the OVUs that were significantly correlated with weeks after injection also had a significant positive correlation with tumor size (Figure 5c). Among 3 of them, 2 were unclassified and 1 was classified as *Brunovirus* (OVU#24213). Interestingly, OVU#200524, which was found consistently enriched in AOM models, was also significantly correlated with both weeks (*Q* < 0.0001) and tumor size (*Q* value 0.0002). Because the abundance of these OVUs increased along with week, tumor size and tumor characteristics in AOM models, these OVUs are associated with tumor growth. Fasta sequences of all the unclassified OVUs with significant correlations are attached in Appendix A. The abundance of OVUs significantly correlated with tumor size and the number of tumors is shown in Appendix A. 

### 2.5. Network Correlations among Viral Genera and Their Association with Tumor Growth 

To determine if any viral genera are associated with tumor growth, we used eLSA to reveal both linear and non-linear associations among classified viruses and tumor characteristics. The results showed that the number of tumors, tumor size, and week were positively correlated with each other (Figure 6). Two viral genera showed a significant positive correlation with week, tumor size, and the number of tumors, namely *Brunovirus* and *Hpunavirus* (Figure 6). *Lubbockvirus* showed a significant negative correlation with tumor growth, although the correlation was weak (Figure 6). *Brunovirus* also correlated with other viral genera, including a negative correlation with *Lagaffevirus* and *Mushuvirus* (Figure 6). Because *Lagaffevirus* and *Mushuvirus* also correlated with a group of other viruses, we can deduce that there are two main groups of viruses in this network. One group shows a positive correlation with tumor growth, whereas the other shows a negative correlation with tumor growth. Details of correlations and their associated normalized LS values are listed in Appendix A. The abundance plots of viral genera that are significantly correlated with tumor growth are provided in Appendix A. 

## 3. Discussion

Recent studies showed that CRC patients have an altered gut virome compared to healthy controls; however, when and how the virome changes remains unknown. To our knowledge, this is the first longitudinal study to analyze the fecal virome during the development of colorectal neoplasia. We showed that gut virome alteration takes place during tumor development and identified potential viruses associated with tumor growth. 

We showed a significantly lower alpha diversity and shift in viral profile in AOM treated mice when tumors first appeared in week 12. Unlike the study by Nakatsu et al., which showed an increase in virome diversity in CRC human patients, we showed a decrease in virome diversity in our mouse model [17]. This inconsistent finding may be due to our study being performed in mice with polyps, instead of in humans with CRC. Nonetheless, our findings indicate that tumor formation is correlated with the gut virome alteration. Interestingly, this virome alteration pattern was not observed in the control mice and only occurred when tumors first appeared. In this regard, this change in diversity is likely a tumor-associated effect rather than an aging effect. 

In addition to a decrease in alpha diversity in AOM treated mice in week 12, a significantly lower alpha diversity (both the observed number of OVUs and Shannon diversity) was also observed in PBS models at week 20, however, this pattern cannot be detected in the AOM models. We hypothesize that this alteration in alpha diversity in PBS models is caused by aging and the injection of AOM accelerates this aging effect, therefore the decrease in alpha diversity was observed earlier (in week 12). Another hypothesis is that our sample size was too small to draw a definitive conclusion. The Shannon diversity at the time points after the significant decrease in both PBS- and AOM-treated mice did not show any significant changes as compared to those at the early time points before the decrease. If our first hypothesis was right, the changes at the later time points after the decrease should remain statistically significant as compared to those at the early time points before the decrease. However, no such changes were observed. This may be due to the large variation of Shannon diversity at the later time points as evidenced by the large error bars (Figure 2b). Thus, the same experiment with a larger sample size should be conducted in the future to examine which hypothesis is correct. 

Given the underexplored virome and the lack of taxonomic information for viruses (which comprises more than 55% of our sequences) [19], virome analysis was first carried out at the OVU level. We identified four OVUs that are correlated with tumor growth, the most important one being OVU# 200524, which shows a significant positive correlation with tumor size and week and is consistently enriched in the AOM virome. However, because this OVU is unclassified, no further information can be deduced without further analysis. OVU#24213 and OVU#73531, which are also positively correlated with tumor growth, are classified as members of the genus *Brunovirus*. *Brunovirus* also shows a positive correlation with tumor growth at genus level correlation and is enriched in week 24 in AOM models, implying its likely role in tumor development. Interestingly, this genus is only present from week 12 onwards in both AOM and PBS models, however, its abundance in AOM models is much higher compared to in PBS models. The reason for this pattern is unknown. In addition to *Brunovirus*, *Hpunavirus* is also positively correlated with tumor growth and enriched in week 24 in AOM models. On the other hand, *Lubbockvirus* is negatively correlated with tumor growth, which suggests that it may play a role in inhibiting tumor growth. All these important viral genera are bacteriophages that belong to the family *Myoviridae*, consistent with findings by Hannigan et al. which showed the virome signature in CRC was driven by OVUs from *Myoviridae* [18]. Because viral genera from the same family show opposite associations with tumor growth, correlation analysis at levels lower than the family level are crucial to identify more reliable associations. While Nakatsu et al. identified important viruses at the genus level, such as *Orthobunyavirus*, *Inovirus*, and *Tunalikevirus*, none of our classified OVUs belong to these viral genera [17]. 

Because this study mainly focused on identifying virome signatures in colorectal neoplasia, we did not assess the interactions between bacteria and bacteriophages. Further investigations including analyzing the interactions among bacteria, bacteriophages, and their mammalian hosts are needed to determine how the mammalian host and the surrounding microbial community are affected by bacteriophages (if any). If specific interactions are found between bacteriophages and bacteria, their interactions need to be further validated by in vitro and in vivo experiments. 

There are several limitations of our study. Firstly, we did not sequence enough timepoints to confirm whether the gut virome change occurred before or after tumor formation. Our existing data show a significant decrease in Shannon diversity and shift in viral profile in AOM models at week 12, however, tumors already formed at week 12. Therefore, we cannot deduce whether the virome change occurred before or after tumor formation without examining other time points. Examining time points before or after tumor formation, such as weeks 11 and 13, will allow us to see when the virome change occurred more precisely. Secondly, the fecal virome may not reflect the mucosal virome. A previous study found that the microbiota of fecal and mucosal samples was significantly different (both paired and non-paired), however, differences between CRC patients and controls were still evident [20]. Analysing the mucosal virome remains technically challenging. Mouse colorectal biopsies obtained from the colonoscopy in live mice or mouse colorectal tissues obtained from groups of mice euthanised at different time points will not be large enough for both the downstream virome extraction and the histological examination when the tumor was very small at the early stages of colorectal cancer development. If these early timepoints were not analysed, the important profile of the mucosal virome during the tumor initiation will be missed. The majority of the fecal DNA (~99%) belongs to the bacteriome and virome, so more complete viral sequences could be easily identified in the fecal DNA. In contrast, DNA extracted from the colorectal tissues for mucosal virome were inherantly and inevitably contaiminated with the mouse DNA which contributed to >90% of the total tissue DNA. This low viral to host DNA ratio hinders the discovery of viral sequences if the same sequencing depth were used as in the fecal DNA sequencing. One solution is to increase the seqeunecing depth to obtain enough sequencing reads after a stringent fitlering of the host DNA sequences but this is not financially feasible at the moment. The other solution is to isolate the virus-like particles (VLPs) from the mouse colorectal tissues before DNA extraction and sequencing. This will be our future study. Thirdly, the virome is different among different mouse species and different sources [21]. Thus, the experiments in this study should be repeated by using mice from different animal facilities and pet stores in future. Alternatively, as our main aim is to understand the relationships between human virome changes and the CRC development, transplantation of human fecal virome from healthy individuals and patients with adenoma or CRC into our mouse model will also be performed to analyse the virome changes along with CRC development in the future. Fourthly, we only sequenced dsDNA, therefore we may miss out on some of the potentially important ssDNA viruses and RNA viruses. Lastly, A/J mice used in this study have been previously shown to be highly susceptible to AOM-induced tumor initiation when compared to other mouse strains [22]. This high susceptibility of A/J mice may be caused by their defective *Naip5* gene [23] which has been shown to protect mice from colonic tumorigenesis [24]. This could explain why we observed spontanesous colorectal tumors in some of the PBS-treated mice. Although the penetrance, multiplicity, and size of these tumors in the PBS-treated mice was significantly lower than those in AOM-treated mice (Figure 1), this may interfere with the virome profile. Thus, it would be crucial to verify our results in mice with a functional *Naip5* gene, such as C57BL6, in the future.

Our study provides strong evidence of an increase in certain bacteriophage abundance along with tumorigenesis, however, as mentioned previously, the interaction between bacteriophage and bacteria has not been assessed. In addition, further functional analysis is needed to determine whether these novel bacteriophages carry genes that can contribute to human biology alteration or direct interaction with human cells. With the developing virome enrichment techniques, shotgun metagenomics of mucosal virome is becoming feasible. Our findings of important OVUs and viral genera in colorectal neoplasia provides future directions for exploring the mucosal virome change along the adenoma-carcinoma sequence and comparing if consistent changes can be observed. The development of bioinformatics tools dedicated to virome analysis, in particular, eukaryotic viruses, is also crucial. Recently, tools such as VIBRANT have been very useful in recovering viruses from mixed microbial sequences. However, it can only identify bacteriophages but not eukaryotic viruses, which on its own is not sufficient to recover viruses for mucosal virome study. Thus, continued effort in improving virome enrichment and extraction techniques, and virome analysis tools are needed to gain a better understanding of the gut virome of colorectal neoplasia and lead to mechanistic insights into how the known viruses and viral dark matter contribute to the development of colorectal neoplasia.

In summary, our analysis indicates that the viral profile changes, along with the development of colorectal neoplasia, may play an important role in promoting colorectal carcinogenesis. This provides insight into the development of novel therapeutics for the prevention and treatment of colorectal cancer.

## 4. Materials and Methods

### 4.1. Study Design and Sample Collection

This animal study (18/153A) was approved by the UNSW Animal Ethics Committee. Sixteen mice were purchased from Animal Resources Centre (Canning Vale WA 6970 Australia) via Biological Resources Centre (BRC). Mice arrived at BRC St George at 12 weeks of age. They were inspected and acclimatized for 7 days following arrival. The animal weighing was repeated once per week. During these weekly weighing of mice, stool samples were collected via defecation. 

After the acclimation period, four A/J mice per sex were intraperitoneally injected with 8mg/kg AOM or PBS weekly for 6 weeks, and the colons of the mice were monitored biweekly by colonoscopy. At least a pair of endoscopically normal and tumor tissues were biopsied during colonoscopies of each mouse at different time points. Tissues were formalin fixed, paraffin embedded, stained with hematoxylin and eosin (H&E), and examined by a pathologist blindly. Stool samples from weeks 0, 10, 12, 14, 20, and 24 were subjected to downstream analysis. The number of tumors was calculated by analysis of colonoscopic videos, and the tumor size was scored based on the diameter of the colon lumen occupied by a tumor, according to a published protocol [25]. 

### 4.2. DNA Extraction and Shotgun Metagenomic Sequencing 

DNA was extracted from the stool samples using PSP^®^ Spin Stool DNA Kit (ThermoFisher, MA, USA, #IVK1038110300) as per the manufacturer’s instructions. Briefly, the stool was mechanically lysed using Qiagen TissueLyser II (Qiagen, MD, USA #85300). Following removal of impurities, DNA was eluted, and concentration was measured with Qubit dsDNA HS Assay Kit (Life Technologies, MA, USA #Q32853) and Qubit Fluorometer (Life Technologies, MA, USA).

Library preparation for shotgun metagenomic sequencing was performed with Nextra DNA Flex Library Prep (Illumina, CA USA, #20018705) as per the manufacturer’s instructions. Libraries were sequenced on NovaSeq 6000 Sequencing System (Illumina, CA USA, #20012850) at the Ramaciotti Centre for Genomics (UNSW, Sydney, Australia). 

### 4.3. Metagenomic Sequence Analysis and Classification 

Bioinformatics analysis was conducted using an in-house bioinformatics pipeline (Appendix A). Duplicated paired-end fastq reads were grouped and de-duplicated using clumpify (parameter dedupe) from the BBMap2 package (https://sourceforge.net/projects/bbmap/ accessed on 12 March 2020). Fastp was used to remove low-quality reads and trim reads with low-quality bases in the end with default parameter [26]. Host contamination was removed by mapping reads to the GRCm38 mouse reference genome using minimap2 [27]. Decontaminated paired-end read files were assembled using MEGAHITv1.2.9 (meta-sensitive pre-set) [28]. All assembled contigs were taken as input by VIBRANT to identify the viral contigs (VCs) [29]. In addition, assembled contigs that were annotated by Kraken2 as viruses were also extracted [30]. VCs ≥ 5 kb from VIBRANT and Kraken 2 were merged. VCs were clustered using CD-HIT-EST to generate OVUs with 95% sequence identity, 90% alignment coverage for the shorter sequence, and 95% length difference cut-off (-c 0.95, -aS 0.9, -s 0.95) [31]. Two tools were used for taxonomy annotation: vConTACT2 and Kraken2 [30,32]. Kraken2 uses a kmer-based approach, whereby each k-mer in the sequence is mapped to the lowest common ancestor of the genomes that contain that k-mer in the reference database. Sequences are classified by querying the database for each k-mer in a sequence and then using the resulting set of lowest common ancestor (LCA) taxa to determine an appropriate label for the sequence [30]. Unlike Kraken2, vConTACT2 performs taxonomic assignment by clustering proteins from user-given genomes and the reference database [32]. For the clustered OVUs, taxonomy information is determined by comparing the viral cluster (VC) and VC subcluster with the reference genome. If the OVU is within the same VC subcluster as the reference genome, then they are likely to be in the same genus. If the OVU is in the same VC but not the same subcluster as a reference, then it is highly likely the two genomes are related at a genus–subfamily level [32]. This method allows the discovery of new viruses; however, it only works for prokaryotic viruses as it uses a pre-build database that only contains prokaryotic viruses or archaeal viruses, and it only assigns up to the genus level. To get a more comprehensive annotation of the virome, results from vConTACT2 and Kraken2 were merged. The “ProkaryoticViralRefSeq201-Merged” database was used for vConTACT2, and was run at default settings [32]. Kraken2 was run using a kraken2 metagenomic virus database [33].

### 4.4. Statistical Analysis 

Raw sequencing reads were mapped to OVUs using Bowtie2 and the abundance of the OVUs within each sample was calculated [34]. OVU read counts were normalized to fragments per kilobase per million (FPKM) mapped reads. OVUs ≥ 10 kb were used in diversity analysis. Alpha diversity was calculated at OVU level with otuSummary [35]; Kruskal–Wallis test, Dunn’s test, and pairwise Wilcoxon test were performed to determine differences between groups. Beta diversity was calculated using the Bray–Curtis metric. Statistical significance was assessed with the PERMANOVA test (PERM = 999). All *p* values were adjusted using Benjamini–Hochberg multiple test correction. *p*-values of <0.05 were considered statistically significant. All diversity calculations were conducted with R package Phyloseq and Vegan package [36,37]. All plots were plotted with R package ggplot2 [38]. 

### 4.5. Differential Abundance, Correlation Analysis, and Network Analysis

To examine whether there are differences in virome signature in AOM and PBS treated mice, linear discriminant analysis (LDA) effect size (LEfSe) was used to identify viruses enriched in these models. Alpha value ≤ 0.05 was considered to be significant and effect size LDA score cut off was set to 2 [39]. Correlation analysis was performed using the R package MaAslin2 to detect any significant correlations between OVUs, time, and tumor characteristics (tumor size and tumor numbers) [40]. Lowly abundant OVUs (FPKM count ≤ 1000) were excluded from correlation analysis with week and tumor size. The Q value threshold for significance was set at 0.05. In addition, extended local similarity analysis (eLSA) was used to reveal both linear and non-linear associations among classified viruses and tumor characteristics [41]. The significance of local similarity (LS) scores was based on 1000 permutations. LS score was standardized by dividing all LS scores by the absolute value of the maximum LS score. Correlations of standardized LS score ≥0.4 and ≤−0.4 (except for virus association with tumor characteristics), and *p*-value ≤ 0.05, Q value ≤ 0.1 was considered significant. ELSA networks were visualized using Cytoscape [42].

## Figures and Tables

**Figure 1 pathogens-11-00457-f001:**
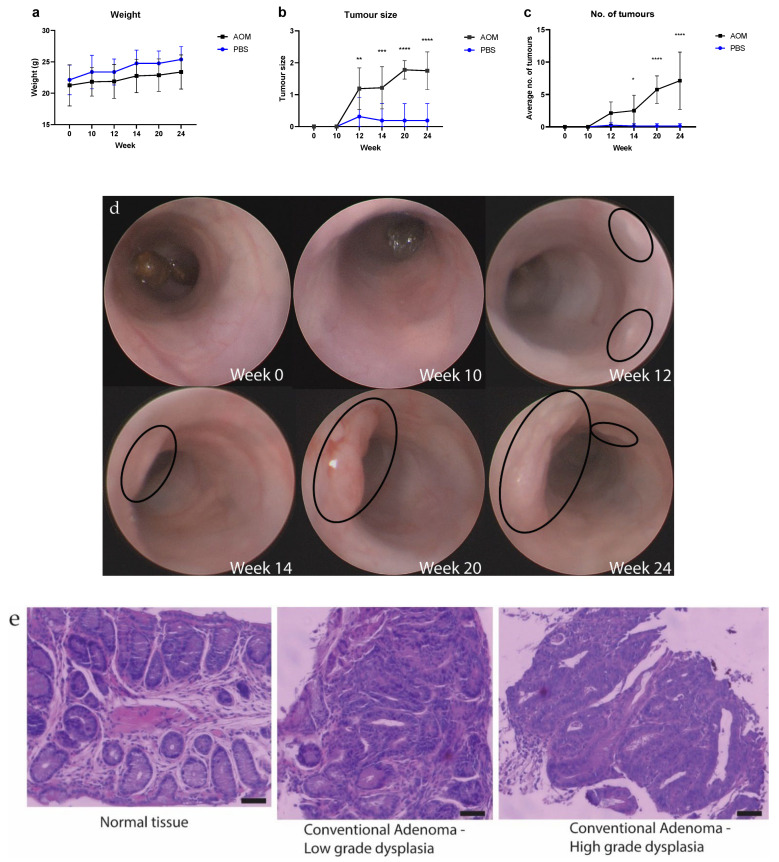
(**a**) Weight (g) of AOM- and PBS-treated mice weeks after the initial injection. (**b**) Mean tumor size in AOM and PBS treated mice at weeks 0, 10, 12, 14, 20, and 24. (**c**) Mean number of tumors in AOM and PBS treated mice at weeks 0, 10, 12, 14, 20, and 24. (* *p* ≤ 0.05; ** *p* ≤ 0.01; *** *p* ≤ 0.001; **** *p* ≤ 0.0001). (**d**) Representative colonoscopic images of AOM-treated mice at indicated weeks after initial injection. Tumors were circled in black. (**e**) Representative H&E images of the colorectal biopsies taken from AOM-treated mice. Scale bar = 50 μm.

**Figure 2 pathogens-11-00457-f002:**
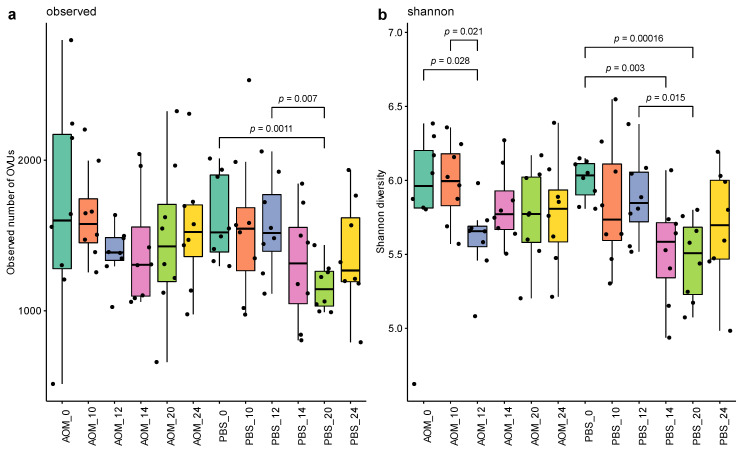
(**a**) Alpha diversity based on the observed number of OVUs according to sample time. (**b**) Alpha diversity based on Shannon diversity. For panels a and b, box plots indicate median (middle line), 25th, 75th percentile (box), *p* was calculated by pairwise Wilcoxon test.

**Figure 3 pathogens-11-00457-f003:**
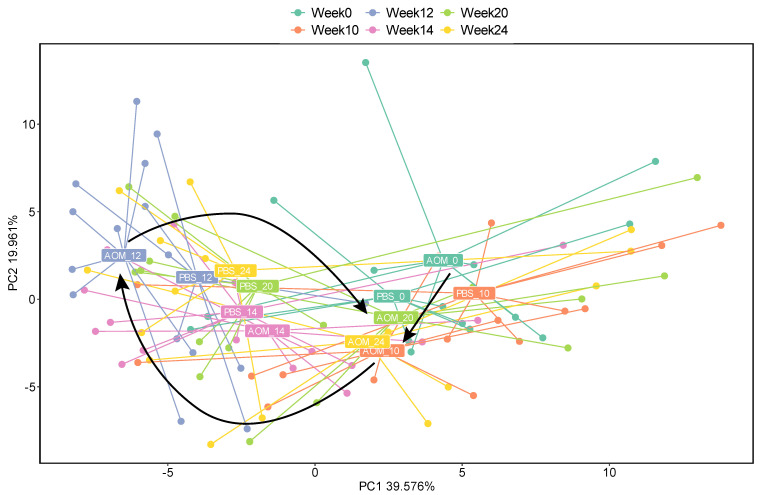
Principle coordinates analysis plot showing the separation of viral profile in PBS and AOM treated mice across weeks. The black arrow shows the shift in the viral profile of AOM virome in weeks 0, 10, and 12.

**Figure 4 pathogens-11-00457-f004:**
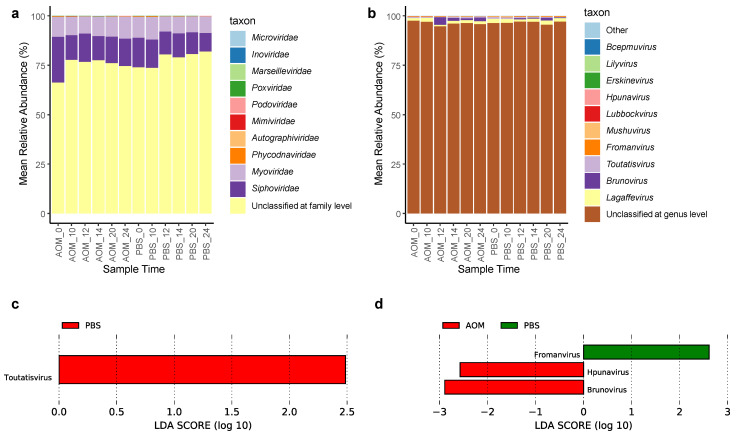
(**a**) Stacked bar plot of viral families with top 11 mean relative abundance (%), other viral families were grouped into other. (**b**) Stacked bar plot of viral genera with top 12 mean relative abundance (%), other viral genera were grouped into other. (**c**) Linear Discriminant Analysis (LDA) scores of differentially abundant viral species in AOM and PBS models (*p* < 0.05, LDA score > 2.0) in week 10. (**d**) LDA scores of differentially abundant viral species in AOM and PBS models (*p* < 0.05, LDA score > 2.0) in week 24. The LDA score indicates the effect size and ranking of each differentially abundant taxon.

**Figure 5 pathogens-11-00457-f005:**
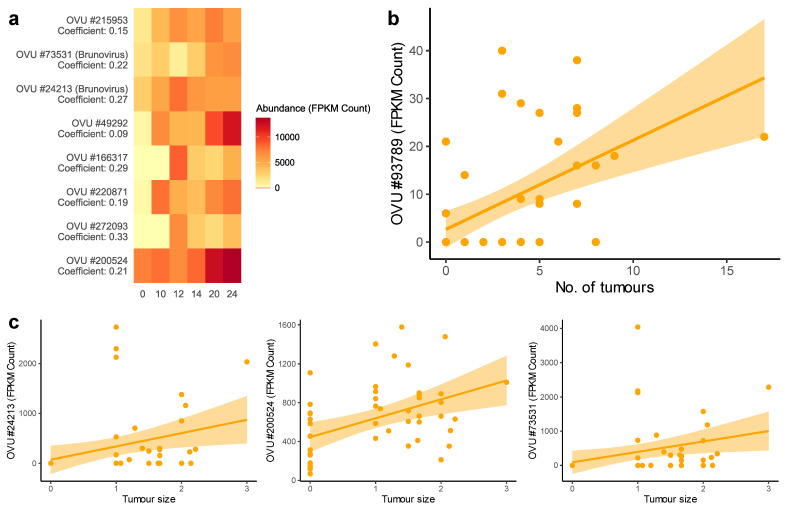
(**a**) Abundance of OVUs significantly correlated with weeks after injection in carcinogen (AOM) treated mice (*n* = 48). *X*-axis represents weeks, *y*-axis represents OVUs with their correlation coefficient. All of the significant correlations have a *Q* value ≤ 0.05. (**b**) Correlation between the abundance of OVU and tumor number in carcinogen (AOM) treated mice (*n* = 48). (**c**) Correlation between the abundance of OVUs with FPKM count ≥ 1000 and tumor size in carcinogen (AOM) treated mice (*n* = 48). For panels b and c, the orange line shows the linear regression line and the orange area shows the 95% confidence interval for the slope.

**Figure 6 pathogens-11-00457-f006:**
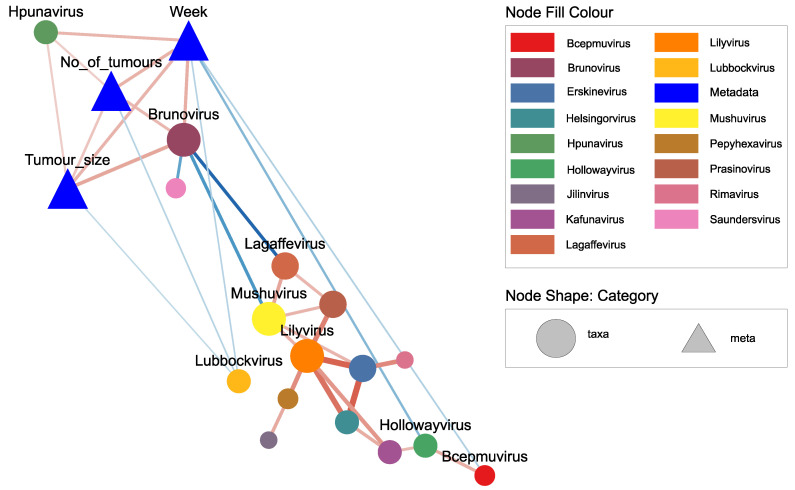
Association network among classified viral species and tumor characteristics. Only statistically significant (*p* ≤ 0.05, *Q* ≤ 0.1) and strong correlations (normalised local similarity (LS) value ≥ 0.4 and ≤−0.4) are shown in the figure (except for the correlation between viral species and tumor characteristics). The size of the nodes is proportional to the degree of the node and the colour represents their corresponding genus. Circular nodes represent viral species, triangles represent metadata. The line thickness is proportional to the absolute value of the normalised LS score. Red lines represent positive correlation and blue lines represent negative correlation.

**Table 1 pathogens-11-00457-t001:** OVUs enriched in AOM and PBS models by LEfSe comparisons.

OVU	Week	LDA Score	*p* Value	Treatment
OVU #208593	0	2.010315	0.034698	AOM
OVU #200524	10	2.238268	0.023742	AOM
OVU #208593	10	2.184009	0.015087	AOM
OVU #200524	12	2.371702	0.003276	AOM
OVU #200524	14	2.217095	0.006323	AOM
OVU #208593	20	2.008747	0.027279	AOM
OVU #200524	24	2.297522	0.011719	AOM
OVU #272093	0	2.596756	0.049141	PBS
OVU #79336	0	2.465271	0.027423	PBS
OVU #175582	0	2.41953	0.02731	PBS
OVU #175582	10	2.498688	0.011719	PBS
OVU #79336	10	2.485104	0.027423	PBS
OVU #272093	12	2.639696	0.045361	PBS
OVU #79336	12	2.520702	0.033006	PBS
OVU #175582	12	2.416228	0.025181	PBS
OVU #272093	14	3.058463	0.035556	PBS
OVU #99458	14	2.42021	0.014214	PBS
OVU #99458	20	2.704742	0.015333	PBS
OVU #99458	24	2.494088	0.030087	PBS

## Data Availability

Sequence data that support the findings of this study have been deposited in National Center for Biotechnology Information (NCBI) with the primary accession code: PRJNA801109.

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
