# Peer review of "Fecal DNA Virome Is Associated with the Development of Colorectal Neoplasia in a Murine Model of Colorectal Cancer"

_pathogens, 2022, doi:10.3390/pathogens11040457_

Round 1

Reviewer 1 Report

Li and colleagues studied a hot topic "fecal virome associated with the development of  colorectal neoplasia in a azoxymethane (AOM)-induced murine model of colorectal cancer. The authors showed that a significantly lower alpha diversity and shift in viral profile when tumors first appeared in week 12,  and  viruses from the genera Brunovirus, Hpunavirus are positively corelated with tumor developemnt and size, especially in at late time points.

Although the topic is important, I have many comments on this paper

1- The authors only detected DNA, not RNA virome. And I see this in the limitation of the study, so the authors need to change the title to reflect this point " Fecal DNA virome", otherwise the readers would expect RNA virome analysis.

2- I wonder why the authors use fecal samples for analysis of the gut virome, especially they have the model. It is worthy to use the colon samples (especially the polyp region) to assess the gut virome. Since this is real virome community associated with tumor. I can understand that Nakatsu et al. Gasteroenterology  used fecal samples due to the difficulty to obtain the colon biopsies samples from the patients. But here you have the model.

3- Importantly, the  virome is different from different mice species and different sources. This point should be added in the discussion and analysis of analysis to compare the virome in the same background https://www.mdpi.com/2076-2607/9/10/2064

4- I am surprised about  data of supple data 1-metadata, especially the no of tumor and tumor size are not matched. I understand they should also match, but also the authors did not provide representative images for polyp development and quantification in this model.

5- Table 1. , Supple table 5 and supple table 7:  The authors need to verify which viruses (family, species) or unclassified are associated with OVU#, few are mentioned in text, while other not. 

6- Is there supple table 6

Author Response

Response to Reviewer 1 Comments

Point 1: The authors only detected DNA, not RNA virome. And I see this in the limitation of the study, so the authors need to change the title to reflect this point " Fecal DNA virome", otherwise the readers would expect RNA virome analysis.

Response 1: We agreed with reviewer 1 and have changed the title to “Fecal DNA virome is associated with the development of colorectal neoplasia in a murine model of colorectal cancer"

Point 2:  I wonder why the authors use fecal samples for analysis of the gut virome, especially they have the model. It is worthy to use the colon samples (especially the polyp region) to assess the gut virome. Since this is real virome community associated with tumor. I can understand that Nakatsu et al. Gasteroenterology  used fecal samples due to the difficulty to obtain the colon biopsies samples from the patients. But here you have the model.

Response 2: We currently encountered two technical problems in analysing the mucosal virome in the mouse colorectal tissues. Firstly, mouse colorectal biopsies obtained from the colonoscopy in live mice or mouse colorectal tissues obtained from groups of mice euthanised at different time points were not large enough for both the downstream virome extraction and the histological examination when the tumor was very small at the early stage of colorectal cancer development. If we ignored these early timepoints, we may miss out the important profile of the mucosal virome during the tumor initiation. Secondly, the majority of the fecal DNA (~99%) belongs to the bacteriome and virome so more complete viral sequences could be easily identified in the fecal DNA as demonstrated in our manuscript. In contrast, DNA extracted from the colorectal tissues was inherently and inevitably contaminated with the mouse DNA which contributed to >90% of the total tissue DNA. This low viral to host DNA ratio hinders the discovery of viral sequences if the same sequencing depths were used as in the fecal DNA sequencing. There are two ways to overcome this shortfall. One solution is to increase the sequencing depth to obtain enough sequencing reads after stringent filtering of the mouse DNA sequences. However, we discovered that this is not a cost-effective way after testing this option. The other solution is to isolate the virus-like particles (VLPs) from the mouse colorectal tissues before DNA extraction. We are currently optimising the condition for isolating VLPs and would like to report the findings in our future report. We have added these discussions to the manuscript.

Point 3: Importantly, the virome is different from different mice species and different sources. This point should be added in the discussion and analysis of analysis to compare the virome in the same background https://www.mdpi.com/2076-2607/9/10/2064

Response 3: We agreed that the virome is different among different mouse species and different sources. Thus, the CRC experiments should be repeated by using mice from different animal facilities and pet stores in the future. In addition, as our main aim is to understand the relationships between human virome changes and CRC development, transplantation of human fecal virome from healthy individuals and patients with adenoma or CRC into our mouse model will also be performed to analyse the virome changes along with the CRC development. This limitation has been added to the discussion section of the manuscript.

Point 4 I am surprised about  data of supple data 1-metadata, especially the no of tumor and tumor size are not matched. I understand they should also match, but also the authors did not provide representative images for polyp development and quantification in this model.

Response 4: The numbers of tumors shown were the total number of tumors per mouse per time point while the tumor sizes shown were the average tumor size per mouse per timepoint. To make it clearer, we provided the individual tumor size per time point in the supplementary data 1-metadata and added representative image for polyp development in Figure 1.

Point 5 - Table 1. , Supple table 5 and supple table 7:  The authors need to verify which viruses (family, species) or unclassified are associated with OVU#, few are mentioned in text, while other not. 

Response 5: In Supplemenatary Table 5, we have added columns of family and genus that each OVU# belongs to. We also added an extra table in Supplementary table 7 that listed all the OVU# belonging to each genus shown.

Point 6 - Is there supple table 6  

Response 6: We are sorry that we didn’t upload the supplementary data 6 which contains the fasta sequence for all the OVUs that show a significant correlation with week, tumour size or tumour number. We have uploaded this data.

Reviewer 2 Report

This manuscript describes the change of characteristics of the gut virome during tumor formation in colorectal cancer. Overall, the manuscript was well-written in good English and met the high quality standard of the journal. The study design is straightforward, the method used is appropriate, the results are well presented and clearly described, and the data interpretation is proper. It is an important study to show how the viral profile had shifted during neoplasia despite most of the findings were not statistically significant.

I list below my comments for improvement.

If aging is the reason for the decreased diversity observed in the PBS-treated mice, shouldn’t the same changes is also shown in the AOM-treated mice? In the discussion, the authors hypothesized that maybe this effect was accelerated to week 12 in the AOM-treated group but can these changes be reversed with following (advancing) age as shown in figure 2?

Were the mice sacrificed at the end of the study? Any pathological examination (histology) on the tumor will be useful.

Author Response

Response to Reviewer 2 Comments

Point 1 If aging is the reason for the decreased diversity observed in the PBS-treated mice, shouldn’t the same changes is also shown in the AOM-treated mice? In the discussion, the authors hypothesized that maybe this effect was accelerated to week 12 in the AOM-treated group but can these changes be reversed with following (advancing) age as shown in figure 2?

Response 1: No significant changes in shannon diversity between PBS- and AOM-treated mice at each paired time point were observed. This suggests that regardless of the age, AOM did not have any effect on shannon diversity. In contrast, a statistically significant decrease in shannon diversity was only observed in the PBS-treated mice at week 20 as compared to those at week 0 and 12, and in the AOM-treated mice at week 12 as compared to those at week 0 and week 10. No significant changes in shannon diversity were observed among other time points in both PBS- and AOM-treated mice. Two possible hypotheses could explain this phenomenon. Firstly, once the PBS-treated mice reached at week 20 or the AOM-treated mice reached at week 12, the shannon diversity decreased and maintained at the lower level until week 24. Thus, this supports our idea that the decrease was accelerated to week 12 in the AOM-treated mice. Another possibility is that our sample size was too small to draw a definitive conclusion. The shannon diversity at the time points after the significant decrease in both PBS- and AOM-treated mice did not show any significant changes as compared to those at the early time points before the decrease. If our first hypothesis were right, the changes at the later time points after the decrease should remain statistically significant as compared to those at the early time points before the decrease. However, no such changes were observed. This may be due to the large variation of Shannon diversity at the later time points as evidenced by the large error bars (Figure 2b). Thus, the same experiment with a larger sample size should be conducted in the future to examine which hypothesis is correct. We have added this alternative explanation in the discussion section of the manuscript. 

Point 2 Were the mice sacrificed at the end of the study? Any pathological examination (histology) on the tumour will be useful.

Response 2: Yes. The mice were sacrificed at the end of the study. No post-mortem pathological examination by histology on the tumor was done. This is because the aim of the study was to study the longitudinal changes of the fecal DNA virome along with the tumor formation. Some tumors appeared and disappeared along this development. The histological examination at the end may not reflect what had happened at the earlier time points. However, we attempted to sample at least a pair of colorectal biopsies from an endoscopically normal tissue and tumor from each mouse at different time points via colonoscopy. Tumors in the AOM-treated mice at week 12 were too small and so no biopsies were taken. Fifty-eight out of 120 (~48.3%) biopsies were lost after the formalin fixed paraffin embedding (FFPE) protocol. This was mainly caused by the small size of these biopsies which were prone to disintegration by the FFPE process. Nevertheless, the remaining 62 (~51.7%) FFPE samples were stained with hematoxylin and eosin (H&E) and were blindly examined by our pathologists. The results of the histological examination were added to the Supplementary Data 1-metadata. Representative H&E images of normal tissues, conventional adenoma with low grade dysplasia and conventional adenoma with high grade dysplasia have been added as Figure1e. These histopathological data confirmed the endoscopic data that the colorectal tissues were correctly classified as normal or tumor.

Round 2

Reviewer 1 Report

The authors addressed my questions.